# The Role of Flaxseed in Improving Human Health

**DOI:** 10.3390/healthcare11030395

**Published:** 2023-01-30

**Authors:** Wioletta Nowak, Małgorzata Jeziorek

**Affiliations:** Department of Dietetics and Bromatology, Faculty of Pharmacy, Wroclaw Medical University, Borowska 211, 50-556 Wroclaw, Poland

**Keywords:** flaxseed, lignans, α-linolenic acid, dietary fiber, bioavailability

## Abstract

Flaxseed contains high amounts of biologically active components such as α-linolenic acid, lignans, and dietary fiber. Due to its numerous nutritional properties, flaxseed has been classified as a “superfood”, that is, a food of natural origin with various bioactive components and many health-promoting benefits. Flaxseed consumption can be an important factor in the prevention of diseases, particularly related to nutrition. The regular consumption of flaxseed may help to improve lipid profile and lower blood pressure, fasting glucose, and insulin resistance index (HOMA-IR). Moreover, flaxseed is characterized by anticancer and antioxidant properties and can significantly reduce the intensity of symptoms associated with menopause, constipation, and mental fatigue, improve skin condition, and accelerate wound healing. In addition to its bioactive compounds, flaxseed also contains antinutrients such as cyanogenic glycosides (CGs), cadmium, trypsin inhibitors, and phytic acid that can reduce the bioavailability of essential nutrients and/or limit its health-promoting effects. Three common forms of flaxseed available for human consumption include whole flaxseed, ground flaxseed, and flaxseed oil. The bioavailability of ALA and lignans is also dependent on the form of flaxseed consumed. To ensure high bioavailability of its bioactive components, flaxseed should be consumed in the ground form.

## 1. Introduction

Since flaxseed contains many essential nutrients, such as lignans, omega-3 fatty acids, dietary fiber, vitamins, and minerals, it has been classified as a “superfood”, that is, a food of natural origin with numerous bioactive components and many health-promoting benefits. Consumption of flaxseed can be an important factor in the prevention of diseases, particularly those related to poor diet [1,2,3].

Flaxseed oil contains about 53% of alpha-linolenic acid (ALA), making it the richest plant source of this compound, and 19% of oleic acid. Flaxseed oil, due to its high content of ALA, has a favorable n-6: n-3 fatty acid ratio of about 0.3:1. Due to its anti-inflammatory and antiproliferative properties, ALA has an anticarcinogenic effect on the human body, thereby preventing the development of malignant tumors and their metastases. Alpha-linolenic acid from flaxseed exerts a positive effect on blood lipids. It has been found to significantly reduce plasma total cholesterol, LDL, and VLDL cholesterol [1,2,4].

Flaxseed is a rich source of dietary fiber (40%), of which the soluble fiber accounts for 25% and insoluble fiber accounts for 75%. A soluble fiber including gums, pectin, and β-glucan plays an important role in reducing glycemia, and absorbing cholesterol and triglycerides, which are key factors in the prevention of cardiovascular disease and diabetes. In addition, soluble fiber effects gut microbiota and may be metabolized to short-chain fatty acid which impacts on human health [4,5,6]. Insoluble fiber consists of cellulose, hemi-cellulose, and lignin, increases the bulk of the stool, and prevents constipation [6]. Flaxseed is a rich source of niacin and vitamin E, particularly in the form of tocopherol, which has strong antioxidant properties [1]. The average tocopherol content of flaxseed ranges from 39.5 to 50 mg/100 g [4]. A proper supply of vitamin E helps to lower the risk of cardiovascular disease, Alzheimer’s disease, and some types of cancer [1].

Flaxseed consists of proteins and peptides which exhibit activities potentially beneficial for human health, such as fungistatic, antihypertensive, antioxidant, and anti-inflammatory activities, and prevents the occurrence of neurodegenerative diseases. Hydrolysates of flaxseed present antidiabetic activity. In addition, flaxseed contains peptides (cyclolinopeptides) which exhibit immunosuppressive, antimalarial, antioxidant, antithrombotic, and antifungal properties [7].

Flaxseed is an excellent dietary source of lignans. After ingestion, lignans in flaxseed are converted by the intestinal microbiota to enterolignans, enterodiol (END), and enterolactone (ENL), which can provide a number of health benefits. The principal lignan precursor found in flaxseed is secoisolariciresinol diglucoside (SDG). Its average content in 100 g of flaxseed is 610–1300 mg. Regular consumption of flaxseed can lower blood pressure, and reduce the risk of dyslipidemia and obesity by lowering body weight and body mass index (BMI) [1,8].

In addition to its bioactive compounds, flaxseed also contains antinutrients such as cyanogenic glycosides (CGs), cadmium, trypsin inhibitors, and phytic acid that can reduce the bioavailability of essential nutrients and/or limit its health-promoting effects [2]. Flaxseed contains 264–354 mg of cyanogenic glycosides per 100 g of seed, including 10–11.8 mg of linamarin/100 g, 136–162 mg of linustatin/100 g, and 105–183 mg of neolinustatin/100 g. Branched-chain amino acids such as valine, leucine, isoleucine, phenylalanine, and tyrosine serve as precursors for CG. The content of CG depends primarily on flaxseed species and the maturity of the seeds. Cyanogenic glycosides are highly toxic, and their high ingestion can pose a risk to human health and life by compromising human nervous, endocrine, and respiratory systems. However, CGs present instability when subjected to thermal processes such as baking, roasting, or boiling, which prevent the formation of hydrocyanic acid responsible for the adverse effects of glycosides. Therefore, mechanical processing such as grinding may effectively inactivate CG [1,2,4,8].

The aim of this review was to present the biological activities of flaxseed and its role in improving human health.

## 2. Forms of Flaxseed Consumption

Three common forms of flaxseed available for human consumption include whole flaxseed, ground flaxseed, and flaxseed oil. The stability of flaxseed and its bioactive components depends on the form of the flaxseed consumed. The ALA in whole flaxseed withstands temperatures up to 350 °C without a negative impact on its oxidative stability. However, grinding the flaxseed breaks the seed coat barrier and increases its sensitivity to oxidation. Therefore, to preserve its health-promoting qualities, whole flaxseed should be stored in airtight containers at room temperature for about 4–20 months. The oil extracted from flaxseed is the most sensitive to oxidation. The important thing is to choose unrefined flaxseed oil which exhibits a longer shelf life due to the presence of antioxidants such as tocopherols and phenolic compounds. Flaxseed oil should be stored refrigerated in an opaque glass container, which prolongs its shelf life for up to 6 months. Flaxseed oil can oxidize even within a week at room temperature and is then unsuitable for consumption. As a functional food ingredient, fatty acids or lignans can be protected against oxidation by adding flaxseed into baked foods, such as muffins or bagels [9,10].

Flaxseed oil contains the highest amount of ALA compared to ground and whole flaxseed. However, due to its short shelf life and lack of stability in unfavorable storage conditions, the recommended form of flaxseed ingestion is ground seeds containing similar ALA levels. Ground flaxseed should be used shortly after being ground and stored for a short period of time, as it is more vulnerable to oxidation than whole flaxseed. Eating whole flaxseed provides the least amount of ALA and SDG due to its hard and impermeable seed coats [9,10,11]. The ingestion of flaxseed oil and ground flaxseed resulted in significantly higher levels of plasma ALA levels compared to whole flaxseed [11].

## 3. Hypocholesterolemic Properties of Flaxseed

Due to its significant content of omega-3 fatty acids, lignans, protein, and soluble fiber, flaxseed may reduce the risk of cardiovascular diseases such as atherosclerosis or ischemic heart disease [12]. Flaxseed ingestion has been linked to improved serum lipid and lipoprotein levels [13]. In the study by Soltanian et al. [13], participants were randomized into three groups. Two intervention groups received either 10 g of flaxseed or psyllium pre-mixed in cookies. The control group received placebo cookies without any additives. Serum lipid and lipoprotein levels were evaluated before the intervention, after 4, 8, and 12 weeks, and 4 weeks after the intervention. Based on the study results, it was observed that 12-week supplementation with flaxseed reduced total cholesterol by an average of 36.9 mg/dL (*p* < 0.001), LDL cholesterol by 21 mg/dL (*p* < 0.001), triglycerides by 12.3 mg/dL (*p* = 0.045), and increased HDL cholesterol by an average of 6.0 mg/dL (*p* = 0.316). Four weeks after the intervention, the authors observed a significant increase in serum total cholesterol (+6.4 mg/dL; *p* < 0.001) and LDL cholesterol (+10.9 mg/dL; *p* = 0.049) in the group supplemented with flaxseed.

A meta-analysis of 14 studies found a correlation between flaxseed ingestion and lipid metabolism parameters in dyslipidemic and healthy participants. In the group of healthy participants, flaxseed supplementation ranged from 2 to 30 g/day and in participants with lipid metabolic disorders it ranged from 15 to 40 g/day. It has been found that flaxseed significantly improved the lipid profile in healthy participants with BMI > 25 kg/m^2^ (SMD: −28.7); (95% CI −54.67–−2.62); (*p* = 0.031), and in dyslipidemic participants (SMD: −1.41); (95% CI −2.30–−0.79); (*p* < 0.001). No significant changes in total cholesterol levels were observed in the control group. Moreover, flaxseed supplementation improved LDL cholesterol (SMD: −0.69); (95% CI −1.13–−0.25); (*p* = 0.002), and reduced triglycerides in dyslipidemic participants (SMD: −1.47); (95% CI −2.21–−0.72); (*p* < 0.001). It also increased HDL cholesterol in healthy (SMD: 5.12); (95% CI 2.34–7.90); (*p* = 0.006) and overweight participants (SMD: 7.92); (95% CI 2.95–12.88); (*p* = 0.002) [14].

Morshedzadeh et al. [15] evaluated the efficacy of flaxseed supplementation in the management of metabolic-syndrome-related parameters, including serum concentration of triglycerides, total cholesterol, and HDL/LDL in patients with mild-to-moderate ulcerative colitis (UC). The study involved 70 participants who were randomized to the intervention group, who received 30 g of ground flaxseed/day for 12 weeks, or to the control group. Serum lipid and lipoprotein levels were evaluated at the beginning and end of the 12-week intervention. The authors observed a significant reduction in the serum concentration of triglycerides (−13.07 ± 8.31 mg/dL; *p* < 0.001) and total cholesterol (−16.50 ± 10.87 mg/dL; *p* < 0.001), and a significant increase in the serum levels of HDL (3.67 ± 2.82 mg/dL; *p* = 0.04) in the group receiving ground flaxseed powder.

Flaxseed supplementation may help to treat dyslipidemia, especially in overweight or obese patients. The form of flaxseed consumed is significantly related to serum lipid and lipoprotein concentrations. Whole flaxseed is more beneficial in regulating lipid metabolism than flaxseed oil [14,16].

## 4. Hypotensive Properties of Flaxseed

Due to its significant content of ALA, lignans, and dietary fiber, flaxseed may be helpful in the treatment of hypertension [17]. The hypotensive effect of flaxseed has been demonstrated in pre-diabetic patients. The study involved 99 pre-diabetic participants divided into three groups: the first and the second groups received, respectively, 40 g and 20 g of flaxseed powder daily for 12 weeks and the third group was the control. Participants’ BP was measured after >10 min of rest, before and after the intervention. The authors observed that, compared to the baseline values (*p* < 0.001), the 40 g group had a significantly lowered SBP (12.24 ± 23.08 mmHg) than the 20 g group (2.56 ± 5.99 mmHg) or the control (−1.5 ± 6.3 mmHg) [18].

A beneficial effect of daily ingestion of flaxseed on SBP and DBP was demonstrated in patients with peripheral artery disease. The study involved 110 patients who were divided into two groups. Every day, the first group ingested a variety of foods that contained 30 g of ground flaxseed daily for over 6 months. The remaining participants were assigned to the control group. Blood pressure was measured at the baseline and after the intervention. The average of a total of three readings was used as the final measurement. Participants had their blood drawn to determine the plasma levels of omega-3 fatty acids and enterolignans, which were used as markers of dietary compliancy. The authors reported a significant reduction in mean SBP (−10 mmHg; *p* = 0.04) and DBP (−7 mmHg; *p* = 0.004) in the flaxseed group compared with the placebo in which SBP rose slightly and DBP remained the same. Patients who had elevated BP at the baseline showed a reduction of 15 mmHg in SBP (*p* = 0.002) and 7 mmHg in DBP (*p* = 0.003) after a 6-month dietary intervention. The plasma levels of ALA and enterolignans increased 2- to 50-fold in the flaxseed-fed group, but did not increase in the placebo [19].

Flaxseed supplementation can significantly reduce BP in adults with hypertension, which was proved in a study involving 112 patients with an age range of 35 to 70 years. Participants were randomized to two groups receiving 10 g and 30 g of flaxseed supplementation for 12 weeks, and one group receiving the placebo. All study participants had their BP measured in a supine position at the baseline, and after 6 and 12 weeks. The authors observed a decrease in SBP (−13.38 mmHg; *p* = 0.001) and DBP (−5.6 mmHg; *p* = 0.001) in the 30 g group compared to the placebo, in which they reported an increase in SBP (1.72 mmHg; *p* = 0.001) and DBP (+2.39 mmHg; *p* = 0.001) [20].

Flaxseed consumption may lower BP, and thus reduce the risk of cardiovascular disease and other diseases in which elevated BP is a major risk factor [17]. However, it is important to consume flaxseed regularly, and the magnitude of BP improvement depends on the consumed portion.

## 5. Hypoglycemic Properties of Flaxseed

Flaxseed supplementation may improve glycemic control and insulin sensitivity in healthy participants and patients with type 2 diabetes [21,22]. The effect of flaxseed consumption on carbohydrate metabolism was evaluated in constipated patients with type 2 diabetes. The study included 77 patients who were randomized into three groups. The first two groups received either 10 g of flaxseed or psyllium pre-mixed in cookies, and the third group received placebo cookies. Fasting glucose and glycosylated hemoglobin were determined at the beginning and end of 4, 8, and 12-week periods. After the 12-week intervention, fasting blood glucose was significantly improved in both the flaxseed (−27.8 mg/dL; *p* = 0.004) and psyllium (−19.7 mg/dL; *p* = 0.004) groups compared to the placebo. However, no significant difference was reported in glycated hemoglobin in either group [13].

Flaxseed supplementation can also play a role in the prevention of type 2 diabetes. The consumption of flaxseed can significantly improve glycemic control, cytokines, and adipokines in overweight and obese participants with pre-diabetes. A 12-week study included 25 overweight or obese patients with pre-diabetes who were randomized into three groups. Participants consumed either 0 g, 13 g, or 26 g of flaxseed per day for 12 weeks. Fasting glucose, insulin, adiponectin, CRP, and HOMA-IR were determined at the beginning and at the end of the study. Fasting glucose (−2 mg/dL; *p* = 0.036), insulin (−1.9 mU/L; *p* = 0.013), and HOMA-IR (−0.6; *p* = 0.008) significantly decreased in the 13 g group compared to the 26 g group and to the placebo. No significant differences were observed in CRP or adiponectin [23].

Another study showed the effect of flaxseed powder on insulin resistance indices and BP in pre-diabetic participants. The study included 99 patients who were randomly divided into three groups. Two groups received 40 g and 20 g of flaxseed powder daily for 12 weeks, and the third group was the control. The authors of the study observed a significant reduction in fasting glucose in all study participants, and the mean of the changes in fasting glucose was not significantly different between groups. Insulin concentration did not change in either group after supplementation with flaxseed. HOMA-IR reduced in the group consuming 20 g of flaxseed (0.27 ± 0.65; *p* = 0.033) compared to the baseline. At the end of the intervention, significant reductions in BP were observed only in participants consuming 40 g of flaxseed (SBP: 12.24 ± 23.08 mmHg; *p* = 0.005; DBP: 4.15 ± 15.55 mmHg; *p* = 0.135) [18].

Flaxseed intake is associated with a significant improvement in carbohydrate metabolism, particularly by lowering fasting glucose. However, further studies with a control group are necessary to determine how regular flaxseed supplementation affects insulin metabolism [21,22,23]. It has been indicated that consuming flaxseed in several smaller portions spread out throughout the day may be more beneficial in reducing glycemia than consuming one large serving at a time [22].

## 6. Anticancer Properties of Flaxseed

As one of the richest plant sources of lignans, flaxseed may reverse early cancerous lesions, and inhibit tumor growth, disease progression, and angiogenesis. Flaxseed lignans demonstrate the ability to interfere with the phenotype of the malignant tumor, which affects its cellular characteristics. In addition, flaxseed lignans, by affecting connections in molecular signaling networks, modulate signaling cascades in various stages of cancer. Therefore, flaxseed lignans can inhibit the progression of various types of cancers [24,25].

Promising anticancerous effects of ENL have been observed in acute myeloid leukemia (AML) cell lines in vitro. Tannous et al. [26] investigated the potential anticancerous effects of flaxseed lignans such as SDG, ENL, and END on AML cell lines in vitro. Acute myeloid leukemia cell lines (KG-1 and Monomac-1) and a normal lymphoblastic cell line were cultured and treated with the purified flaxseed lignans. The authors evaluated cell cycle, cell proliferation, apoptosis, protein expression, and cytochrome c release through cytosolic and mitochondrial fractionation, as well as reactive oxygen species (ROS). They observed that ENL showed a significant selective and dose- and time-dependent cytotoxic effect in both AML cell lines, contrary to normal cells. After 24 h, 48 h, and 72 h, the percent proliferation of the cells treated with 100 µM of ENL significantly decreased, reaching 55% (*p* < 0.0001), 46% (*p* < 0.0001), and 29% (*p* < 0.0001), respectively, in the KG-1 cell line and 55% (*p* < 0.0001), 46% (*p* < 0.0001), and 40% (*p* < 0.0001), respectively, in the Monomac-1 cell line. The cytotoxic effects of ENL were mainly attributed to apoptosis induction and an increase in cellular and DNA fragmentation. In addition, protein expression analysis using Western blots confirmed the activation of the intrinsic apoptotic pathway upon ENL treatment, which was also accompanied by an increase in ROS production intracellularly.

The anticancer effects of flaxseed oil were also demonstrated by studying its direct effects on cancer cell growth in vitro. The treatment of a variety of cancer cell lines with flaxseed oil reduced their growth in a dose-dependent manner, while non-malignant cell lines showed minimal increases in cell growth. B16-BL6 cells were treated with 0.3% or 0.9% flaxseed oil for 4 days. Treatment with 0.3% flaxseed oil decreased the number of cells by about 50% and treatment with 0.9% flaxseed oil completely inhibited B16-BL6 cell growth. The authors of the study observed that cancer cells treated with a 10^−5^ or 10^−6^ M mixture of fatty acids, including ALA, DHA, EPA, and the lignans (ENL and END) reduced cell growth [27].

Linoorbitides are another component of flaxseed that exhibit anticancer activity. They are naturally occurring, thermostable, hydrophobic cyclopeptides that exhibit antioxidant effects in flaxseed oil and also have potentials of antitumor activity. Zou et al. [28] evaluated an antiproliferative effect of two flaxseed orbitides ([1–9-NαC], linusorb B2, and [1–9-NαC], linusorb B3) with different structures and they examined their underlying mechanisms of the involvement in the AKT/MAPKs pathway using gastric SGC-7901 cells. In their study, cell viability was analyzed, cell cycle was assessed, and Western blot analysis of cell-cycle-related proteins was performed. The authors of the study found that the two linoorbitides could dose-dependently induce cell cycle arrest of SGC-7901 cells in the G1 phase, accompanied with the downregulation of CDK2, CDK4, cyclin D3, and cyclin E, as well as upregulation of p21WAF1/CIP1 and p27KIP1 analyzed via Western blot. In addition, the authors observed that the disruption of both JNK and AKT played a pivotal role in the [1–9-NαC]-linusorb B2-induced G1 phase arrest, but JNK was the only one involved in the [1–9-NαC]-linusorb B3-induced G1 phase arrest.

Okinyo-Owiti et al. [29] evaluated the cytotoxicity of linoorbitides (flaxseed compounds) with anticancer and antioxidant activity against human breast cancer (Sk-Br-3 and MCF-7) cell lines and the melanoma (A375) cell line. All cell lines were divided into two groups and cultured for 24 h and 48 h. After the specified time, breast cancer and melanoma cells were treated with different concentrations of linoorbitides. The authors of the study observed that the cytotoxicity of linoorbitides against cancer cells was cell-type-specific and concentration-dependent. The authors concluded that the antioxidative activity of linoorbitides may be important in eliciting their cytotoxic effects against cancer cells, which retain high levels of ROS.

Di et al. [25] investigated the effect of the combination of flaxseed lignans (SDG and ENL) and metformin with classic chemotherapeutic agents (docetaxel, doxorubicin, and carboplatin) on the cytotoxic effect of such drugs in the metastatic breast cancer cell lines (SKBR-3 and MDA-MB-231). Each cell line was divided into two groups. The first group was treated with a combination of chemotherapeutic agents and flaxseed lignans for 72 h, and the second group was treated only with chemotherapeutic agents. The authors of the study found that flaxseed lignans significantly enhanced the cytotoxicity of chemotherapeutic agents against breast cancer cells. The study found that the combination of ENL and metformin together in combination with low concentrations of chemotherapeutic drugs was more effective in decreasing cancer cell viability compared to the individual chemotherapeutic drug alone.

## 7. Anti-Inflammatory Properties of Flaxseed

Flaxseed and flaxseed oil are important dietary sources of antioxidants due to their content of lignans, phenolic acids, and tocopherols [30,31]. Flaxseed can lower the concentration of inflammatory markers. Mirfatahi et al. [32] investigated the effects of flaxseed oil consumption on serum systemic and vascular inflammation markers in 34 hemodialysis patients, who were randomly assigned to either the flaxseed oil or the control group. The patients in the flaxseed oil group received 6 g of flaxseed oil daily for 8 weeks, and the control group received 6 g of medium-chain fatty acids (59.4% caprylic acid, 39.6% capric acid, 0.7% caproic acid, 0.2% lauric acid, and 0.1% myristic acid). All patients had their blood drawn at baseline and at the end of Week 8 to determine serum concentrations of high-sensitive C-reactive protein (hs-CRP), soluble intercellular adhesion molecule type 1 (sICAM-1), soluble vascular cell adhesion molecule type 1 (sVCAM-1), sE-selectin, and malondialdehyde (MDA). The authors observed significantly reduced levels of hs-CRP (−1.4 ± 0.5 mg/L; *p* = 0.05) and sVCAM-1 (−23.0 ± 10.0 mg/mL; *p* = 0.05) in the flaxseed oil group at the end of Week 8 compared to baseline and to the control. Moreover, there were no significant differences between the two groups in mean serum concentrations of sICAM-1, sE-selectin, and MDA.

Linusorbs included in the cyclolinopeptides are a component of flaxseed responsible for anti-inflammatory properties. Ratan et al. [33] evaluated the effects of a linusorb mixture on inflammation. In their study, several in vitro (i.e., nitrogen oxide—NO, production, real-time PCR analysis, Western blot analysis) and in vivo analyses with animal inflammation models were conducted. The authors observed that a linusorb mixture inhibited NO production, the cell shape changed, and the inflammatory gene was expressed in stimulated RAW264.7 cells through the direct targeting of Src and Syk in the NF-κB pathway. In addition, the study results showed that a linusorb mixture alleviated symptoms of gastritis, colitis, and hepatitis in murine model systems mediated by the inhibition of Src and Syk.

## 8. The Effect of Flaxseed on Sex Hormones and Menopause Symptoms

Lignans, similar in structure to endogenous estrogens, may influence the concentration of sex hormones in women [34]. Chang et al. [34] assessed the effect of flaxseed on sex hormones, including free estradiol, estradiol, estrone, estrone sulfate, estriol, 2-methoxyestrone, 2-hydroxyestrone, 16α-hydroxyestrone, and the 2:16α-hydroxyestrone ratio. The study was conducted on 99 postmenopausal women, who were randomly assigned to either the intervention or the control group. The intervention group consumed two tablespoons (15 g) of ground flaxseed daily for 7 weeks. Women in the control group maintained their usual diet. The authors found that women in the intervention group had significant increases in total enterolignans (62.3 mg/mL; *p* < 0.001), serum 2-hydroxyestrone concentration (TER: 1.54 pg/mL; 95% CI: 1.18–2.00; *p* = 0.002), and the 2:16α-hydroxyestrone ratio (TER: 1.54; 95% CI: 1.15–2.06; *p* = 0.004) compared to the control. In addition, change in enterolignan level was positively correlated with changes in 2-hydroxyestrone and the 2:16α-hydroxyestrone ratio, and negatively correlated with prolactin levels.

Phytoestrogen-rich flaxseed may decrease menopausal symptoms [35], as it showed beneficial effects on the frequency and intensity of hot flashes [36].

The study evaluating the effects of flaxseed on menopausal symptoms involved 90 menopausal women, who were randomly assigned to one of three groups. Women in the first group received 1 g daily of flaxseed extract containing at least 100 mg of SDG. Women in the second group received 90 g daily of ground flaxseed containing at least 270 mg of SDG. The third group received 1 g of collagen daily (the control group). Endometrial thickness, vaginal cytology, and the severity of menopausal symptoms were assessed using the Kupperman index before and 6 months after the intervention. The authors observed that in the group supplementing flaxseed extract or ground flaxseed, the intensity of symptoms associated with menopause significantly decreased compared to the control group. In the first group, the Kupperman index and hot flashes were reduced by 2.5 (*p* = 0.007) and 1.6 (*p* = 0.001), respectively, and in the second group they were reduced by 3.05 (*p* = 0.005) and 1.04 (*p* = 0.035). However, no significant effects were observed on the vaginal epithelium or endometrium in both intervention groups [37].

Another study analyzed the effect of flaxseed on menopausal symptoms in 140 women who were divided into four groups. Women in the first group did not use hormone replacement therapy and received 5 g of flaxseed daily, and women in the second group used hormone replacement therapy and received 5 g of flaxseed daily. Women in the third group used only hormone replacement therapy, and the fourth group was the control. Before and after the study, participants were asked to complete a patient assessment questionnaire and answer questions about the severity of their menopausal symptoms and quality of life. The authors reported significant decreases in menopausal symptoms and increases in quality of life in women consuming flaxseed. The intensity of menopausal symptoms decreased by 8.7% and 9.8% in two intervention groups (*p* < 0.05) [35].

## 9. Flaxseed and Digestive Health

The ingestion of 0.3 g of flaxseed for each kg of body weight per day during 1 week by adult men triggered a significant increase in ENL blood concentration, accompanied by fecal excretion of propionate and glycerol. In addition, ENL production was linked to the abundance of *Ruminococcus bromii* and *Ruminococcus lactaris* [38]. Flaxseed is a rich source of dietary fiber and can be used to prevent and treat constipation. The consumption of 10 g of flaxseed twice a day for 12 weeks significantly improved constipation symptoms, particularly stool consistency [39].

Flaxseed, due to the content of soluble fiber, may have a beneficial effect on gut microbiota. In a study conducted by Brahe et al. [40], they investigated the effect of dietary interventions with *Lactobacillus paracasei* F19 or flaxseed mucilage (10 g) on gut microbiota and metabolic risk markers in obesity. The intake of flaxseed mucilage over 6 weeks led to a reduction in serum C-peptide and insulin release (*p* < 0.05) and improved insulin sensitivity (*p* < 0,05). After a 6-week intervention with flaxseed mucilage, they observed alterations in the abundance of thirty-three metagenomic species (*p* < 0.01), including a decreased relative abundance of eight *Faecalibacterium* species.

Flaxseed oil also plays a role in constipation prevention. The ingestion of 6.9 ± 2.7 mL of flaxseed oil per day for 4 weeks improved the Rome III score significantly (*p* < 0.01). The supplementation of flaxseed oil mainly contributed to increasing the frequency of evacuation and improving the consistency of stools [41].

Daily consumption of flaxseed contributes to increasing fecal fat excretion. It may play an important role in the prevention of being overweight and of obesity. The study conducted by Kristensen et al. [42], which was completed by 16 participants, proved that the consumption of 5 g/day of viscous dietary fibers from flaxseeds for one week significantly increased fecal excretion of fat and energy. The participants with flaxseed fiber drinks excreted 4.96 ± 0.31 g/d of fat, while participants with flaxseed fiber bread eliminated 3.76 ± 0.31 g/d of fat with feces (*p* < 0.001).

Sant’Ana et al. [43] investigated the effect of brown and golden flaxseeds on intestinal permeability and endotoxemia of perimenopausal overweight women. In the group of participants with supplementation of 40 g of brown or golden flaxseeds for 12 weeks, a significant reduction in intestinal permeability was observed, with the delta of the lactulose/mannitol ratio being smaller (*p* ≤ 0.05).

To conclude, the consumption of flaxseed contributes to a reduction in intestinal permeability and endotoxemia.

## 10. Flaxseed and the Nervous System

The consumption of flaxseed may have beneficial effects on mental fatigue and the nervous system. Gholami et al. [44] assessed the effect of flaxseed on physical and mental fatigue in children and adolescents with excess body weight using the Multidimensional Fatigue Inventory, the Short Mood and Feeling Questionnaire, Depression Anxiety Stress Scales, Council on Nutrition Appetite Questionnaire, and Pittsburgh Sleep Quality Index. The study included 72 participants with a BMI > 25 kg/m^2^ who were randomized into two groups. The first group consumed 20 g of flaxseed and the second group consumed 25 g of puffed wheat for 4 weeks. Fatigue, mood feelings (anxiety, stress, depression), and appetite were measured, as well as height, waist circumference, and body weight. The authors observed a significant reduction in mental fatigue in the group consuming flaxseed compared to the group receiving puffed wheat. However, regular consumption of flaxseed did not affect general fatigue, motivation, or activity, nor did it affect depression or anxiety.

Flaxseed oil supplementation may be related to brain-derived neurotrophic factor (BDNF) and the psychological status of women with depression. The study participants were randomly divided into two groups. The first group consumed a 1000 mg flaxseed oil capsule twice a day for 10 weeks, while the other consumed a placebo. Serum BDNF and anthropometric measurements were taken before and after the intervention. The intensity of depression symptoms was assessed using the Beck Depression Inventory-II (BDI-II) questionnaire. The authors of the study found that serum BDNF concentration increased significantly (1.12 ± 0.6 pg/mL vs. 0.2 ± 0.56 pg/mL; *p* < 0.001) and the total BDI-II score was significantly lower (−16.62 ± 7.03 vs. −8.45 ± 7.8; *p* < 0.001) in the supplementation group compared with the placebo. Depression symptoms were reduced or eliminated in the study participants being supplemented with flaxseed oil [45].

## 11. Flaxseed and the Skin

Omega-3 fatty acids play an important role in skin physiology, and it has been shown that flaxseed oil supplementation can significantly improve skin condition. In the study by Neukam et al. [46], participants were divided into two groups. The first group received four capsules of flaxseed oil (555.32 mg/capsule) and the second group received four capsules of safflower oil (560 mg/capsule) for 12 weeks. Plasma polyunsaturated fatty acids, skin sensitivity, skin hydration, and transepidermal water loss were evaluated at the baseline and at Weeks 6 and 12. Supplementation with flaxseed oil led to significant decreases in sensitivity, skin roughness, scaling, and transepidermal water loss, while epidermal hydration and smoothness were increased compared to the group being supplemented with safflower oil [46].

Flaxseed omega-3 fatty acids may improve wound healing. Supplementation with 1000 mg omega-3 fatty acids from flaxseed oil supplements twice a day for 12 weeks was shown to significantly reduce the length and depth of ulcers in patients with diabetic foot syndrome compared to the control group [47].

The biological activities of flaxseed and its health-promoting effects mentioned in the study are presented in Table 1.

## 12. Conclusions

Regular flaxseed consumption improves lipid profile by lowering total cholesterol, LDL cholesterol, and triglycerides, and by increasing HDL cholesterol in healthy people with excess body weight and in patients with dyslipidemia. This may be crucial in the prevention of many diseases, particularly cardiovascular diseases. Whole flaxseed is more beneficial in regulating lipid metabolism than flaxseed oil. Flaxseed supplementation can significantly reduce SBP and DBP, and thus prevent the development of diseases in which elevated BP is an important risk factor. The hypotensive effects of flaxseed are proportional to its serving size. Flaxseed can modulate carbohydrate metabolism by lowering fasting glucose levels and the HOMA-IR, thereby preventing the onset of type 2 diabetes or insulin resistance. However, the researchers’ observations suggest that these health benefits can be obtained by consuming flaxseed in smaller portions and at regular intervals throughout the day. Since flaxseed can increase cell apoptosis and inhibit tumor growth, it may be effective in preventing the onset of cancer and inhibiting or limiting its progression, as well as increasing the cytotoxicity of chemotherapeutic drugs used in cancer patients. Flaxseed has antioxidant properties as it increases total antioxidant capacity and reduces serum MDA levels. The consumption of flaxseed can significantly reduce the intensity of menopausal symptoms. Due to its high fiber content, flaxseed can be used to reduce symptoms of constipation. Moreover, flaxseed can help with mental fatigue and depression symptoms. Regular flaxseed consumption has a beneficial effect on skin condition by increasing its hydration, reducing roughness, sensitivity, and transepidermal water loss, and accelerating the wound healing process.

To get the most health benefits out of flaxseed, it should be consumed in the right form. The most beneficial form is ground seeds as they provide significant amounts of ALA and SDG. It is important to consume it freshly ground and in the shortest possible time. Thermal processing does not adversely affect the content of ALA and SDG; therefore, they can be eaten raw and thermally treated. However, heating linseed oil is not recommended.

Flaxseed is versatile, easy, and cheap to include in a human diet. It may be a natural form of supplementation of many essential nutrients. From the review of the scientific literature, it is possible to unequivocally confirm the effectiveness of flaxseed supplementation in improving human health. Further studies are needed to clearly establish the recommended portion of flaxseed consumption to bring into effect its biological activities.

## Figures and Tables

**Table 1 healthcare-11-00395-t001:** The biological activities of flaxseed and its health-promoting effects.

Biological Activities and the Related Diseases	Compounds Responsible for Biological Activity	Form of Flaxseed	Design of Clinical Trials	Results of the Clinical Trials
**Improvement in lipid profile;** **decreased risk of heart diseases**	α-linolenic acid, phenolic compounds, lignans, dietary fiber	Flaxseed pre-mixed in cookies	77 participants; 12-week intervention Intervention group: 10 g of flaxseed or psyllium pre-mixed in cookies per day; control group: cookies without any additives	Reduced total cholesterol level (36.9 mg/dL; *p* < 0.001), LDL-C (21 mg/dL; *p* < 0.001), and TG (12.3 mg/dL; *p* = 0.045); improved HDL-C (6.0 mg/dL; *p* = 0.316) in the flaxseed group [13]
		Flaxseed oil, roasted flaxseed, ground flaxseed, raw flaxseed	In the group of healthy participants: 2 to 30 g/d of flaxseed; in the group of participants with lipid metabolic disorders: 15 to 40 g per day of flaxseed	Improvement in lipid profile in healthy participants with BMI >25 kg/m^2^ (SMD: −28.7); (95% CI −54.67–−2.62); (*p* = 0.031), and in dyslipidemic participants (SMD: −1.41); (95% CI −2.30–−0.79); (*p* < 0.001); improvement in LDL-C (SMD: −0.69); (95% CI −1.13–−0.25); (*p* = 0.002), and reduced TG in dyslipidemic participants (SMD: −1.47); (95% CI −2. 21–−0.72); (*p* < 0.001); increased HDL-C in healthy (SMD: 5.12); (95% CI 2.34–7.90); (*p* = 0.006) and overweight participants (SMD: 7.92); (95% CI 2.95–12.88); (*p* = 0.002) with flaxseed supplementation [14]
		Ground flaxseed	70 participants; 12-week intervention Intervention group: 30 g of ground flaxseed per day; control group: normal diet without flaxseed supplementation	Reduction in TG (−13.07 ± 8.31 mg/dL; *p* < 0.001), and total cholesterol (−16.50 ± 10.87 mg/dL; *p* < 0.001) and increase in HDL-C (3.67 ± 2.82 mg/dL; *p* = 0.04) in the flaxseed group [15]
**Hypotensive properties;** **decreased risk of hypertension and other cardiovascular conditions**	α-linolenic acid, lignans, dietary fiber	Flaxseed powder	99 participants; 12-week intervention with a control group; intervention groups: 20 or 40 g of flaxseed powder per day	The 40 g group had a significantly lowered SBP (12.24 ± 23.08 mmHg) compared to the 20 g group (2.56 ± 5.99 mmHg) and the control (−1.5 ± 6.3 mmHg) [18]
		Ground flaxseed	110 participants; 6-month intervention with a control group; intervention group: 30 g of ground flaxseed per day	Significant reduction in SBP (−10 mmHg; *p* = 0.04) and DBP (−7 mmHg; *p* = 0.004) in the flaxseed group; patients who had elevated BP at the baseline: reduction of 15 mmHg in SBP (*p* = 0.002) and 7 mmHg in DBP (*p* = 0.003) [19]
		Flaxseed powder	112 participants with hypertension; 12-week intervention with a control group; intervention groups: 10 or 30 g of flaxseed per day	Decrease in SBP (−13.38 mmHg; *p* = 0.001) and DBP (−5.6 mmHg; *p* = 0.001) in the 30 g group [20]
**Hypoglycemic properties;** **decreased risk of type 2 diabetes and insulin resistance**	Dietary fiber, α-linolenic acid	Flaxseed pre-mixed in cookies	77 participants; 12-week intervention; intervention groups: 10 g of flaxseed or psyllium pre-mixed in cookies per day; control group received placebo cookies without any additives	Significantly improved fasting glucose (−27.8 mg/dL; *p* = 0.004) in the flaxseed group [13]
		Flaxseed powder	99 participants; 12-week intervention with a control group; intervention groups: 20 or 40 g of flaxseed powder per day	Reduced HOMA-IR in the group consuming 20 g of flaxseed (0.27 ± 0.65; *p* = 0.033); significant reduction in fasting glucose in all study participants [18]
		Flaxseed powder	25 participants overweight or obese with pre-diabetes; 12-week study; participants consumed 0 g, 13 g, or 26 g of flaxseed powder per day	Fasting glucose (−2 mg/dL; *p* = 0.036), insulin (−1.9 mU/L; *p* = 0.013), and HOMA-IR (−0.6; *p* = 0.008) significantly decreased in the 13 g group compared to the 26 g group and placebo [23]
**Anticancer properties**	Lignans, linoorbitides, α-linolenic acids	-	Breast cancer cell lines (SKBR-3 and MDA-MB-231) were divided into two groups; the first was treated with the combination of chemotherapeutic agents and flaxseed lignans for 72 h, and the second was treated only with chemotherapeutic agents	Flaxseed lignans significantly enhanced the cytotoxicity of chemotherapeutic agents against breast cancer cells; the combination of ENL and metformin together in combination with low concentrations of chemotherapeutic drugs was more effective in decreasing cancer cell viability compared to the individual chemotherapeutic drug alone [25]
		-	Acute myeloid leukemia cell lines and a normal lymphoblastic cell line were cultured and treated with various concentrations of the purified flaxseed lignans	After 24 h, 48 h, and 72 h, the percent proliferation of the cells treated with 100 µM of ENL significantly decreased, reaching 55% (*p* < 0.0001), 46% (*p* < 0.0001), and 29% (*p* < 0.0001) in the KG-1 cell line, respectively; and 55% (*p* < 0.0001), 46% (*p* < 0.0001), and 40% (*p* < 0.0001) in the Monomac-1 cell line, respectively; ENL induces apoptosis and increases cellular and DNA fragmentation [26]
		-	Treatment of a variety of cancer cell lines with 0.3% or 0.9% flaxseed oil for 4–6 days	Cancer cells treated with a 10^−5^ or 10^−6^ M mixture of fatty acids and the lignans reduced cell growth; treatment with 0.3% flaxseed oil decreased the number of cells by about 50% and treatment with 0.9% flaxseed oil completely inhibited B16-BL6 cell growth [27]
		-	SGC-7901 cells were treated with various concentrations of [1–9-NαC]-linusorb B2 or B3 (80, 120, and 200 μM) for 24 h and subjected to flow cytometric analysis to examine the DNA content after PI staining; the expression levels of CDK2, CDK4, cyclin D3, cyclin E, and p27KIP1 were measured using Western blotting assay	Cell population in the G1 phase of the cell cycle increased from 33.53 ± 1.46% to 35.30 ± 1.59%, 40.03 ± 2.33%, and 46.30 ± 1.45% after treatment with [1–9-NαC]-linusorb B3 of 80, 120, and 200 μM, respectively; the percentage in G1 phase reached 43.50 ± 2.05%, 49.96 ± 1.90%, and 56.45 ± 0.72%, following exposure to [1–9-NαC]-linusorb B2 of 80, 120, and 200 μM, respectively; downregulation of CDK2, CDK4, cyclin D3, and cyclin E, as well as upregulation of p21WAF1/CIP1 and p27KIP1; the disruption of both JNK and AKT played a pivotal role in [1–9-NαC]-linusorb B2-induced G1 phase arrest, but only JNK was involved in [1–9-NαC]-linusorb B3-induce G1 phase arrest [28]
		-	Human breast cancer cell lines and melanoma cell lines were divided into two groups and cultured for 24 h and 48 h; then, were treated with different concentrations of linoorbitides	Cytotoxicity of linoorbitides against cancer cells was cell-type-specific and concentration-dependent [29]
**Anti-inflammatory properties**	Lignans, phenolic acids, tocopherols, linusorbs	Flaxseed oil	34 participants with hemodialysis; 8-week intervention; intervention group: 6 g of flaxseed oil per day Control group: 6 g of medium-chain fatty acids (59.4% caprylic acid, 39.6% capric acid, 0.7% caproic acid, 0.2% lauric acid, and 0.1% myristic acid)	Reduction in hs-CRP (−1.4 ± 0.5 mg/L; *p* = 0.05) and sVCAM-1 (−23.0 ± 10.0 mg/mL; *p* = 0.05) levels in the flaxseed oil group [32]
		-	RAW264.7 and HEK293 cells were treated with either LOMIX (0–200 g/mL), L-NAME (0–1 mM), Pred (0–100 µM), or individual linusorb; then, several in vitro assays (i.e., NO production, real-time PCR analysis, Western blot analysis) and in vivo analyses were carried out	Inhibition of NO production, cell shape changes, and inflammatory gene expression in stimulated RAW264.7 cells through direct targeting of Src and Syk in the NF-κB pathway; alleviation of symptoms of gastritis, colitis, and hepatitis in murine model systems mediated via inhibition of Src I Syk [33]
**Influence on the concentration of female sex hormones; reduced risk of breast cancer**	Lignans	Ground flaxseed	99 postmenopausal women; 7-week intervention; intervention group: 2 tablespoons (15 g) of ground flaxseed daily Control group: women on a normal diet	Significant increase in total enterolignans (62.3 mg/mL; *p* < 0.001), serum 2-hydroxyestrone concentration (TER: 1.54 pg/mL; 95% CI: 1.18–2.00; *p* = 0.002), and 2:16α-hydroxyestrone ratio (TER: 1.54; 95% CI: 1.15–2.06; *p* = 0.004); change in enterolignan level was positively correlated with changes in 2-hydroxyestrone and 2:16α-hydroxyestrone ratio, and negatively correlated with prolactin levels [34]
**Reduction in menopausal symptoms**	Lignans	No data available	140 menopausal women; 3-month intervention with a control group; intervention groups: the first group received 5 g of flaxseed daily, the second group received hormone replacement therapy and 5 g of flaxseed daily, and the third group used hormone replacement therapy	Significant decrease in menopausal symptoms and increase in quality of life in women consuming flaxseed. The intensity of menopausal symptoms decreased by 8.7% and 9.8% in two intervention groups (*p* < 0.05) being supplemented with flaxseed [35]
		Flaxseed extract, ground flaxseed	90 menopausal women; 6-month intervention; intervention groups: 1 g of flaxseed extract containing at least 100 mg of SDG, or 90 g of ground flaxseed containing at least 270 mg of SDG per day Control group: 1 g of collagen daily	In the flaxseed extract group, Kupperman index and hot flashes were reduced by 2.5 (*p* = 0.007) and 1.6 (*p* = 0.001), respectively, and in the ground flaxseed group by 3.05 (*p* = 0.005) and 1.04 (*p* = 0.035) compared to the control group [37]
**Regulation of gut microbiota**	Lignans, soluble fiber	Ground flaxseed	9 healthy men; 1-week intervention; each participant ingested 0.3 g of flaxseed for each kg of body weight per day	Significant increase in ENL blood concentration, accompanied by fecal excretion of propionate and glycerol; ENL production was linked to the abundance of *Ruminococcus bromii* and *Ruminococcus lactaris* [38]
		Flaxseed mucilage	58 obese postmenopausal women; 6-week intervention with a control group; intervention groups: daily intake of *L. paracasei* F19 or 10 g of flaxseed mucilage	Alterations in abundance of thirty-three metagenomic species (*p* < 0.01), including decreased relative abundance of eight *Faecalibacterium* species in the flaxseed mucilage group [40]
**Increase in the frequency of evacuation and improved consistency of stools; prevention of constipation**	Dietary fiber, flaxseed oil	Baked flaxseed	53 constipated patients with T2D with BMI 20.5–48.9 kg/m^2^; 12-week intervention; 10 g of flaxseed pre-mixed in cookies twice per day or placebo cookies for 12 weeks	Significantly improved constipation symptoms, particularly stool consistency [39]
		Flaxseed oil	50 constipated patients; 4-week intervention; intervention groups: the mineral oil group, the olive oil group, and the flaxseed oil group The initial oil dose was 4 mL/day, and adjustments during the follow-up could be made as needed. The average consumption of flaxseed oil was 6.9 ± 2.7 mL per day	The Rome III score improved significantly in patients receiving mineral oil (10.5 ± 5.0 to 4.1 ± 4.0; *p* < 0.01), olive oil (10.3 ± 4.2 to 3.2 ± 3.8; *p* < 0.01), and flaxseed oil (9.6 ± 4.2 to 6.0 ± 5.1; *p* < 0.01), with no significant group-by-time interaction (*p* < 0.15). The scores of 5 from 6 constipation symptoms reduced similarly in the mineral oil and olive oil groups, whereas only the frequency of evacuation and the consistency of stools improved in the flaxseed oil group [41]
**Increased fecal fat excretion; prevention of being overweight and of obesity**	Soluble dietary fiber	Flaxseed fiber drink, flaxseed fiber bread	17 participants; one-week intervention; intervention groups: a diet with flaxseed fiber drink or with flaxseed fiber bread Control group: low-fiber diet	Participants drinking flaxseed fiber drinks excreted 4.96 ± 0.31 g/d of fat, as compared to only 3.20 ± 0.33 g fat/d with the control diet, corresponding to a 55% increase. Participants eating flaxseed fiber bread eliminated 3.76 ± 0.31 g/d of fat with feces (*p* < 0.001) [42]
**Reduction in mental fatigue**	α-linolenic acid	Uncooked ground flaxseed	72 children and adolescents with a BMI > 25 kg/m^2^; 4-week intervention; intervention group: 20 g of flaxseed daily Control group: 25 g of puffed wheat daily	Significant reduction in mental fatigue in the group consuming flaxseed [44]
**Reduction in or elimination of depression symptoms**	α-linolenic acid	Flaxseed oil	60 depressed women; 10-week intervention with a control group; intervention group: 1000 mg of flaxseed oil capsule twice a day	Increase in serum BDNF concentration (1.12 ± 0.6 pg/mL vs. 0.2 ± 0.56 pg/mL; *p* < 0.001) and decrease in total BDI-II score (−16.62 ± 7.03 vs. −8.45 ± 7.8; *p* < 0.001) in the intervention group [45]
**Improvement in skin condition**	α-linolenic acid	Flaxseed oil	26 women with sensitive skin; 12-week intervention; intervention group: 4 capsules of flaxseed oil (555,32 mg/capsule) per day Control group: 4 capsules of safflower oil (560 mg/capsule) per day	Significant decrease in sensitivity, skin roughness, scaling, and transepidermal water loss, while epidermal hydration and smoothness were increased in the intervention group [46]
**Improvement in wound healing**	Omega-3 fatty acids	Flaxseed oil	60 participants with grade 3 diabetic foot syndrome; 12-week intervention; intervention group: 1000 mg of omega-3 fatty acids from flaxseed oil twice a day	Significant reduction in the length and depth of ulcers in the intervention group [47]

LDL-C—low-density lipoprotein cholesterol; TG—triglicerydes; HDL-C—high-density lipoprotein cholesterol; BMI—body mass index; SBP—systolic blood pressure; DBP—diastolic blood pressure; HOMA-IR—homeostatic model assessment for insulin resistance; ENL—enterolactone; hs-CRP—high-sensitive C-reactive protein; sVCAM-1—soluble vascular cell adhesion molecule type; NO—nitrogen oxide; PCR—polymerase chain reaction; T2D—type 2 diabetes; BDNF—brain-derived neurotrophic factor; BDI-II—Beck Depression Inventory-II.

## Data Availability

Not applicable.

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
