# Peer review of "The Role of Flaxseed in Improving Human Health"

_healthcare, 2023, doi:10.3390/healthcare11030395_

Round 1

Reviewer 1 Report

The presented article is valuable and interesting. However my notices are listed below:

1. The title seems to be completelly unfortunate. It requires reconsideration to be clear. I have serious doubts whether flaxseed promotes anything.

2. The statements are confused: "However, CG present instability when subjected to thermal processes such as baking, roasting or boiling, which prevent the formation of hydrocyanic acid responsible for the adverse effects of glycosides. Therefore, mechanical processing such as grinding, may effectively inactivate CG". Is the griding thermal process?

3. In sentence: "The aim of this study was to present the results indicating the role of flaxseed consumption and human health" the main message is unclear and does not match the title.

4. In general, most international organizations, e.g. WHO, prefer the use of Arabic numerals rather than Roman numerals in the nomenclature regarding types of diabetes.

5. The final conclusion: "However, to get the most health benefits out of flaxseed, it should be consumed in the right form. The most beneficial are ground seeds as they provide significant amounts of ALA and SDG. It is important to consume it freshly ground and in the shortest possible time"

Is it possible to explain if it is raw or heat treated?

6. Is it possible to present some important information in the form of a table? This could improve the attractiveness of the article.

Author Response

Dear Reviewer

Thank you for your recognition of the value of our work. We appreciate the effort the reviewers have made and believe that their suggestions will improve our manuscript. We hope that our replies and changes to the manuscript will be satisfactory.

Reviewer 2 Report

In general, I find this review interesting. The results of relevant clinical trials related to the corresponding types of human diseases are summarized in this work. I also appreciate that the authors highlight the most appropriate form of flaxseed for the highest bioavailability of the contained bioactive compounds. The structure of the manuscript is well organized.

However, some parts and topics of this review must be necessarily extended or specified since it needs to cover some important human health-promoting effects of flaxseed constituents sufficiently.

The comments and suggestions related to single parts of the review which should improve the quality of your manuscript are stated below:

·       The “Introduction” generally describes the biological activities of flaxseed oil, dietary fiber and lignans. Concerning dietary fiber, in the manuscript should be described differences in biological activities between soluble and insoluble dietary fiber. Since the insoluble fiber act as a bulk laxative, mainly with the potential to treat constipation, the soluble fiber in the form of flaxseed mucilage possesses more properties like antioxidant, anti-obesity, anti-cholesterol and anti-diabetic activities. Also, flaxseed proteins and naturally occurring peptides exhibit activities potentially beneficial for human health, such as fungistatic (proteins) and antioxidant, immunosuppressive, antimalarial, antithrombic and antifungal (peptides) (https://doi.org/10.3390/foods11152304).

·       In the chapter “Anticancer properties of flaxseed” is reported cytotoxicity of linoorbitides as the flaxseed compounds. Authors should specify that linoorbitides (a.k.a. cyclolinopeptides, linusorbs) are a group of naturally occurring, thermostable hydrophobic cyclopeptides and may exhibit quite a broad scale of biological activities. Except for the activity against breast cancer and melanoma cell lines, it was reported the suppression of the growth of gastric carcinoma cells (https://doi.org/10.1016/j.jff.2018.11.002). Flaxseed cyclolinopeptides are also studied for their immunosuppressive and anti-inflammatory antidiabetic and cardio-protective activities. If they exist, the authors should report more references related to the health-beneficial effects of cyclolinopeptides supported by clinical trials in the relevant chapters of the manuscript.

·       The chapter “Flaxseed and the digestive health” narrowly focuses on the effect of flaxseed consumption on the presence of enterolignans. However, flaxseed may also possess many other digestive health-promoting effects. Flaxseed exhibits prebiotic capacity and may modulate gut microbiota (e.g. https://doi.org/10.1017/S0007114515001786). The insoluble fiber of flaxseed may act as a bulk laxative, thus similarly to flaxseed oil may be used to treat constipation (e.g. https://doi.org/10.1053/j.jrn.2014.07.009). Flaxseed and involved compounds may also have other health-beneficial effects confirmed by clinical trials. They increase fat excretion (e.g. https://doi.org/10.1186/1743-7075-9-8), reduce intestinal permeability and possess preventive effects against endotoxemia (e.g. https://doi.org/10.1080/09637486.2022.2052820). Based on the versatile potential of flaxseed within digestive health, I suggest extending this chapter and concluding the significant findings of clinical trials.

·       I suggest preparing a table where the important biological activities of flaxseed and its health-promoting effects will be summarized. This table could contain the form of flaxseed, the compound(s) responsible for certain biological activity (if it is known), biological activities and the related disease(s), design of clinical trial(s) (if they are available) - optional, results of the clinical trial(s) and the relevant references.

·       Finally, in the chapter “Conclusions”, the authors could report the current state of the applications of bioactive compounds from flaxseed as the active ingredients in pharmaceuticals. I also would like to see authors give their own opinions and make suggestions for future research about flaxseed and its health-promoting compounds in treating human diseases.

Author Response

(The authors gave the same response as above.)

Round 2

Reviewer 2 Report

The manuscript has been revised adequately to my commentaries and suggestions thus I recommend it for publication in Healthcare.